# Title Cross-Sectional Study to Evaluate Knowledge and Attitudes on Oral Hygiene of Romanian Students

**DOI:** 10.3390/medicina58030406

**Published:** 2022-03-09

**Authors:** Catalina Iulia Saveanu, Cosmin Constantin Cretu, Irina Bamboi, Alexandra Ecaterina Săveanu, Daniela Anistoroaei

**Affiliations:** Department of Dent-Alveolar and Maxillo-Facial Surgery, Faculty of Dental Medicine, University of Medicine and Pharmacy Grigore T Popa, 700115 Iasi, Romania; cisaveanu@prevod.umfiasi.ro (C.I.S.); daniulia05@yahoo.com (D.A.)

**Keywords:** oral hygiene, toothbrushing, toothpaste, dental floss

## Abstract

*Background and Objectives:* the purpose of this study was to evaluate students’ level of knowledge and attitude towards oral hygiene. *Materials and Methods:* the evaluation was carried out by a questionnaire, with 30 Q (questions) as follows: demographic data (Q1–Q5), oral hygiene knowledge data (Q6–Q23) and oral hygiene attitude data (Q24–Q30). The study included students from Romanian schools and the selection of the study group was made following selection criteria in accordance with ethical issues. A descriptive statistical analysis was performed and a value of *p* ≤ 0.05 was considered statistically significant. *Results:* the study included a number of 718 subjects with a mean age of 14.54 (±2.22), male 250 (34.8%) and female 468 (65.2%), MS (middle school students) 354 (49.4%) and HH (high school students) 364 (50.6%). Most of the subjects 292 (MS = 160; HS = 132) know a toothbrushing technique, *p* = 0.009, r = 0.091 and 587 (MS = 278; HS = 309) know that brushing removes the bacterial plaque *p* = 0.027, r = −0.082 but only 147 (MS = 71; HS = 76) know that (by) brushing can re-mineralize hard dental structures. The duration of the toothbrushing is variable, for 2- or 3-min *p* = 0.058, r = 0.043. Criteria for choosing the toothbrush were based mainly on the indications of the dentist, respectively, for toothpaste on its properties. The frequency of toothbrushing is mainly twice a day 428 (MS = 234; HS = 248), *p* = 0.079, r = 0.037, 73 (MS = 33; HS = 40) after every meal. *p* = 0.099, r = 0.095. Mouthwash is used by 421 (MS = 199; HS = 222) *p* = 0.111, r = −0.048, and 228 (MS = 199; HS = 222) after each brushing. Dental floss is used by 240 (MS = 106; HS = 134), *p* = 0.031, r = −0.073 and only 74 (MS = 41; HS = 33) after each brushing. *Conclusions:* there are differences in the level of knowledge and attitudes regarding the determinants of oral hygiene depending on the level of education.

## 1. Introduction

Prevention is becoming increasingly important because most diseases that affect the oral cavity can be deterred by appropriate prevention measures. Generally, understanding the factors that influence oral health can help dental professionals to implement an effective strategy. The Center of Disease Control on Oral Prevention recommendations require a sustained education about food hygiene, oral hygiene as well as the importance of both general and local fluoridation, and sealing [1,2,3,4]. The long-term implications of oral health will be found in adult oral health, a health issue that affects both carious lesions and periodontal lesions or even cancer [5,6,7,8,9,10,11]. The results of specialized studies have highlighted the need to establish educational methods designed to improve the determinants of oral health [12,13,14,15]. In addition, numerous studies address the socio-economic and behavioral aspects of children and adolescents in order to highlight their knowledge, attitudes and practices regarding oral health [10,16,17].

Oral hygiene is found in the primary prevention recommendations of the World Health Organization (WHO). In order to change certain behaviors and attitudes, oral hygiene must be evaluated on different population categories, as well as in different regions [6]. In conformity with the results obtained, dental professionals will establish an appropriate therapy that can aim to improve the parameters of oral health. Oral health is a key indicator of overall health, well-being and quality of life. According to WHO reports, more than 530 million children suffer from dental caries of primary teeth and severe periodontal disease, almost 10% of the global population being affected. The Global Burden of Disease Study 2017 estimated that oral diseases affect 3.5 billion people worldwide [18]. According to WHO specifications, improving oral health requires a reform of oral health systems to shift the focus from invasive dental treatment to prevention and as many minimally invasive treatments as possible. WHO has identified key strategies for improving oral health, including prevention through education.

Oral health education aims to inform and develop, among the population, a concept and a hygienic behavior, in order to defend general health, dental and periodontal health, harmonious development and strengthening the body, its adaptation to environmental conditions. Health education can be defined as the sum of all the influences that, together, determine knowledge, concepts and behaviors related to the promotion, maintenance and recovery of health individually and collectively.

These influences include the formative education in the family, school and society, respectively, in the special context of the activity of the health services.

Health education is a communication activity, aimed at improving health and preventing or reducing disease, individually or collectively, by influencing their conceptions, attitudes and behavior, with the help of power and community.

Bacterial plaque is the determining factor in the appearance of tooth decay and periodontal disease. In this context, poor hygiene due to lack of knowledge can lead to compromising the integrity of the tooth. The choice of a toothbrush taking into account its characteristics and a toothpaste taking into account its properties are also very important aspects reported in other specialized studies [19,20,21]. Fluoride is a key agent in reducing the prevalence and severity of dental caries [21,22,23,24,25].

Brushing twice a day with fluoride toothpaste is more efficient, as it maintains adequate fluoride around the teeth for a greater part of the day [26]. In addition, in many countries toothbrushing is part of school routines aimed at improving health [21,25]. Community programs aiming to improve the determinants of oral health should also include nurses from school health services or support staff who can initiate toothbrushing exercises. Thus, these personnel must include oral health education in their regular activities. Teachers, with appropriate guidance, can also motivate and guide students in their toothbrushing. These aspects have long been highlighted in a specialized guide [27]. In this context, oral hygiene is a continuous concern for researchers in the field and must be performed properly by each individual daily at least twice a day [12,13,14,15,16,17].

The negative economic impact through lack of resources will affect the individual from this point of view, but lack of knowledge will affect him even more. Even if, in some situations, the aspects of achieving oral hygiene and the long-term implications in its absence are known, there will always be a number of subjects not sensibilized from this point of view.

The null hypothesis for this study is that there are no differences in the level of knowledge and attitudes of the students regarding oral hygiene depending on the level of education. 

The testable hypothesis was that there are differences in the level of knowledge and attitudes towards oral hygiene depending on the level of education.

The study aimed to evaluate the level of knowledge of students between 10–19 years old regarding oral hygiene. The objectives of this study were to assess the level of knowledge of the students about: toothbrushing technique, choosing the type of toothbrush, choosing toothpaste, the attitudes towards toothbrushing conditions, the use of mouthwash and dental floss.

## 2. Materials and Methods

### 2.1. Study Design and Setting

This is a cross-sectional survey designed to monitor the level of knowledge and attitudes of the student population. The applied questionnaire followed the formative level of the students in terms of oral hygiene. This can help in the future in the design of education programs. This cross-sectional study was carried out according to the formal approval of the research center of the University of Medicine and Pharmacy. For the purpose of the study, seven areas (Iasi, Botosani, Suceava, Prahova, Neamt, Bucuresti, Bacau) of the Romanian region were listed. The state schools were considered in the study based on ease of access.

### 2.2. Study Sample

A preliminary semi-structured questionnaire was originally developed in Romanian translated into English by a professional translator and translated back into Romanian to ensure accuracy. The questionnaire was iteratively tested, with both English and Romanian speakers, to assess the length of the questions, the respondent’s understanding of the questions, to follow the relevance and order of the questions. The final changes to the questionnaire were translated into Romanian by a bilingual staff member and independently reviewed by two other bilingual employees. Many topics were included to understand oral hygiene knowledge and attitudes about students’ daily oral hygiene habits. So, this study included students from 5th to 12th (aged between 10–19 years) grade male and female gender randomly selected from state schools. Sample size estimation was based on the alpha error probability = 0.05, power = 0.95. Thus for 614,767, students in high school education for an error of 5% the sample size is 384 students. The chosen sample is representative for Romania. Our study included 718 students. Survey participants sampling was unlikely. The selection of the study group was made following the selection criteria. The inclusion criteria were as follows: schools where teachers received the information and agreed to distribute the student questionnaire; students who agreed to complete the questionnaire; middle school or high school students. The exclusion criteria were as follows: schools in which teachers did not receive information to send the questionnaire to students; parents who did not agree to have the child complete the questionnaire; problems with the teenager’s desire not to complete the questionnaire; students from another level of study. Completion recommendations were sent to the class guidance teacher and the parents were completely informed by the teachers, through meetings. The agreement to complete the questionnaire was supported by the class teacher, the person under whose supervision this action was possible. The teacher explained it to the parents during the class work sessions. Subjects considered eligible were those who wished to complete this questionnaire after reading its contents.

### 2.3. Study Instrument Development and Validation

The level of knowledge and attitudes of the students regarding oral hygiene was assessed. The questionnaire method was applied for this evaluation. The questionnaire was evaluated by a panel of experts from the Faculty of Dentistry, following a qualitative pre-testing of the content, followed by its validation. The questionnaire was pilot tested with a sample of fifty students to ensure that it was brief and straightforward. In addition, the questionnaire was field-tested to determine its ease of use and accuracy in knowledge items.

### 2.4. Questionnaire Contents

The questionnaire consisted of 30 multiple choice questions with a single correct answer to each question. The oral hygiene knowledge and attitudes of the students were assessed using a structured, questionnaire openly applied and uploaded online on the google docs platform. The questionnaire items included demographic information (Q1–Q5), followed by questions on the level of knowledge of oral hygiene (Q6–Q23) and data on the attitude of the students regarding oral hygiene (Q24–Q30) Table 1.

### 2.5. Assessment of the Oral Hygiene Knowledge and Attitudes

All the multiple-choice questions had a single best response. Every correct answer in the questionnaire received a score of zero, while every incorrect response received a score of one.

### 2.6. Statistical Analysis

The data was collected and introduced into a database. Descriptive statistics of frequency distribution, percentages, and mean knowledge scores were calculated for oral hygiene education. A descriptive statistic of the study was performed by applying crosstabs to all the aspects analyzed according to MS and HH. The processing of statistical data was performed with the program SPSS version 26.00 for Windows, (IBM, Armonk, NY, USA) establishing a threshold of statistical significance of *p* ≤ 0.05. The development of the codebook was based on codes recorded in the interview guide. The codes were grouped into preliminary topics and discussed with the research team to reach a thematic consensus. A member of the team with experience and expertise in qualitative research methods and oral health also reviewed all transcripts, codes and final thematic interpretation. The Chi-square test was used for the comparative analysis in function by study level. The correlation of overall knowledge and attitudes between students was performed using Pearson’s correlation test.

## 3. Results

### 3.1. Demographic Data

The study included a number of 718 students with a mean age of 14.54 (±2.22) the youngest being 10 years old and the oldest being 19 years old, male 250 (34,8%) and female 468 (65.2%), middle school students 354 (49.4%) and high school 364 (50.6%). The distribution of subjects according to class, gender and county is presented in Table 2.

### 3.2. Criteria for Choosing Toothbrush and Toothpaste

The answers regarding the questions related to the criteria used when choosing the toothbrush were in descending order depending on: the indications of the dentist; by age; the design of the toothbrush; the manufacturing company; the price and advertisements Table 3.

The answers to the questions related to the criteria when choosing a tube of toothpaste were in descending order according to: its properties; the manufacturing company; the amount of fluoride; the price; design and advertising Table 3.

### 3.3. The Attitude towards the Amount of Toothpaste Used, the Frequency of Toothbrushing Use

Regarding the question on how much toothpaste you use when brushing your teeth, an approximately equal number of students apply toothpaste over its entire length of the brush 373 (MS = 178; HS = 195) or as much as a pea 340 (MS = 173; HS = 167). *p* = 0.0625, r = −0.035. The frequency of toothbrushing is variable, with most students performing it twice a day for 428 (MS = 234; HS = 248), after each meal for 53 (MS = 25; HS = 28) and once a day for 92 (MS = 55; HS = 37) with *p* = 0.079, establishing a very low positive correlation depending on the level of study r = 0.037. As a time of day, most of the subjects brush their teeth both in the morning and in the evening 508 (MS = 246; HS = 262) and only a small number of subjects, respectively, 73 (MS = 33; HS = 40) brush their teeth after every meal. From this point of view, the results are not significant *p* = 0.099 and correlate very poorly depending on the level of study r = 0.095 Table 4.

Most of the subjects 292 (MS = 160; HS = 132) know a toothbrushing technique, *p* = 0.009, r = 0.091. In addition, the majority 587 (MS = 278; HS = 309), know that brushing removes the bacterial plaque *p* = 0.027, r = −0.082 and the remaining food on the dental surfaces 521 (MS = 264; HS = 257), *p* = 0.233, r = 0.045. About half of the 353 subjects (MS = 183; HS = 170) consider that by applying the brushing they will have whiter teeth *p* = 0.181, r = 0.050 and 147 (MS = 71; HS = 76) know that brushing can remineralize hard dental structures. 

### 3.4. The Attitude towards the Time of Toothbrushing

The duration of the toothbrushing is variable, for most of the students being 2 min for 335 (MS = 182; HS = 153) of the students and 3 min for 209 of the students (MS = 89; HS = 120), *p* = 0.058, r = 0.043 of the students (Figure 1).

### 3.5. The Attitude towards the Mouthwash and Dental Floss

More than half of the students use mouthwash 421 (MS = 199; HS = 222) *p* = 0.111, r = −0.048, after each brushing 228 of the students (MS = 199; HS = 222), followed by the situation when the students feel the need 139 (MS = 56; HS = 83), then by those who use the mouthwash in the morning 42 (MS = 20; HS = 22) and in the evening 31 (MS = 19; HS = 12). Dental floss is used by about a third of students 240 (MS = 106; HS = 134), *p* = 0.031, r = −0.073. Of these, 132 students (MS = 52; HS = 80) use it when they feel the need, 74 (MS = 41; HS = 33) after each brushing, and the rest in the evening, *p* = 0.091, r = −0.038 Table 5.

## 4. Discussion

There are numerous studies in the specialized literature that analyze the relationship between the level of knowledge of the students and the attitude towards oral health [28,29,30,31,32,33]. This study aimed at students’ level of knowledge about oral hygiene. According to Zhu et al., preventive measures are more effective than curative measures [34]. Thus, the information obtained about the students’ level of knowledge will contribute to the improvement of oral health education programs. As female subjects had a higher share in the study 65.18% (468) we did not follow the comparative analysis of knowledge by gender [35]. Oral hygiene is a determining factor in general health [31,36]. According to a specialized study, the number of people with untreated oral conditions increased from 2.5 billion in 1990 to 3.5 billion in 2015 [37]. Therefore, the primary prevention of dental caries should be a priority for both the specialist and the health decision-makers [38,39,40]. The World Health Organization supports the promotion of oral health through educating students [33,39]. Thus, individualized education regarding oral hygiene can provide better premises for oral health, a fact supported by other specialized studies [41,42,43,44]. Other studies show that primary prevention cannot be achieved only by implementing knowledge of oral hygiene with diet being an equally important factor to consider [45,46]. The results of a study show that training courses are needed for hygienic nurses in terms of knowledge about brushing. In this context, the need for educational programs designed to develop oral hygiene skills among students should be emphasized once again.

The most important factor to consider when choosing a toothbrush, for 51.81% of students, was the opinion of a dental professional. The fact that dental professionals are the main source of information about oral hygiene issues is confirmed in other specialized studies [47,48,49]. Other studies suggest that the media is the main source of information [50,51,52]. Choosing the electric brushing technique is an option to achieve better results of oral hygiene monitoring indices [53]. Although 76.60% of the subjects know that when choosing the toothpaste they must take into account its properties, they do not correlate this aspect with the remineralization capacity of the toothpaste a fact highlighted in another specialized study [54]. Only about 20.47% of MH and HH students know about the beneficial effect of fluoride in toothpaste and its role in mineralizing hard dental tissues. The fact that students do not know the properties of fluoride and its beneficial effects has also been highlighted by other authors in the literature [48,49,55]. One study found that the proportion of Australian preschoolers using non-fluoridated toothpaste was higher than in other world regions [56]. A Cochrane Review study supports the benefits of using fluoride toothpaste in preventing tooth decay when compared to non-fluoride toothpaste because a dose-response effect was observed for D (M) FS in children and adolescents [57]. The attitude towards the amount of toothpaste used is generally based on the recommendation to use a pea-sized amount of toothpaste. These is the optimal amount of toothpaste which is considered to be the best, in terms of reduced risk of ingestion and fluoride benefit [58]. Using a larger amount of toothpaste is not so important, the recommendations are to us a pea size amount According to Hu S et al., it is important to control the amount of toothpaste used in order to reduce the risk of fluorosis [59]. The results of a meta-analysis showed that using as much pea as a pea will minimize the risk of fluorosis in children while maximizing the caries-prevention benefit for all age groups [60].

The frequency of brushing is known by only 67.1% of students. The fact that a significant percentage of the students do not know the amount of toothpaste and the frequency with which the brushing should be performed is also highlighted in other specialized studies [48,50,51,52,54,55,61]. The fact that more than half of the students brush their teeth twice a day is beneficial, but it is necessary to improve the level of knowledge about the frequency of oral hygiene, as well as the time of day when it should be performed. In addition, 73 students declared that they brush after each meal as a time of day, but in terms of frequency, only 53 answered that they brush after each meal. This can lead to some variations and lack of credibility regarding the students’ answers. The results of a meta-analysis study showed that the frequency of brushing between more than 2 times a day and less than twice a day does not influence the incidence of carious lesions in general, but they pointed out that incidence and increment of carious lesions was higher in deciduous (OR: 1.75; 95% CI: 1.49 to 2.06) than permanent dentition (OR: 1.39; 95% CI: 1.29 to 1.49) [62]. These results have been highlighted in other specialized studies [63,64,65,66]. Another study showed that the strongest evidence related to caries in the 12-year-old group was found in the frequency of toothbrushing and dental plaque. A study that analyzed the oral health status of adolescents in Shandong province highlighted the importance of visiting a dental practitioner and performing regular oral hygiene to prevent dental caries and gingivitis [67]. The same aspect has been highlighted in other studies that have analyzed the risk factors for dental caries and periodontal disease [68]. 

Knowing a brushing technique is the first step in increasing the efficiency of oral hygiene. The results of our study indicate that most of the subjects 292 (MS = 160; HS = 132) know a toothbrushing technique. A study on the efficiency of the brushing technique and the importance of knowing a brushing technique, pointed out that the Bass brushing technique is much more efficient both in terms of removing bacterial plaque and in terms of maintaining the verticality of the brush bristles for a while longer time [69]. Globally, dental caries and periodontal disease still have a high prevalence rate [70]. Bacterial plaque is one of the determining factors present both in the etiology of carious lesions and in the etiology of periodontal disease [47,71]. Our study shows that about half of the students know a brushing technique, but 81.75% know that brushing removes bacterial plaque, the results obtained being similar to the results of other specialized studies [34,50,54]. In order to receive individualized information on oral hygiene, it is recommended to take into account the provision of access to specialized dental services [44,72].

The attitude towards the mouthwash and dental floss was presented in many studies [73,74,75,76,77,78,79,80,81,82,83,84,85,86,87,88,89,90,91,92,93,94,95]. The use of toothbrush adjuvants is necessary to remove plaque from inaccessible spaces. A study that looked at whether or not motivational interviewing promotes oral health in adolescents found that prevailing health education was less effective than motivational interviewing in evoking favorable changes in the oral health patterns of adolescents and preventing dental caries [73].

The fact that 31.75% of the subjects use mouthwash after brushing is a good attitude, but because only 20.47% of the students know the remineralization capacity, we can say that the difference of 11.28% of the subjects cannot select a mouthwash with remineralization potential. One study found that the use of mouthwash into the size of the caries-preventive effect is less clear [74]. Although some studies do not show the effect of mouthwash in the prevention of tooth decay, the results of a review study that included 37 trials involving 15,813 children and adolescents found that supervised regular use of oral fluoride by children and adolescents is associated with a large reduction in tooth decay of permanent teeth [75]. In addition, the effect of reducing the bacterial plaque of chlorhexidine rinsing solutions is still highlighted in studies, chlorhexidine plays a key role in dentistry and is used to treat or prevent periodontal disease, and has earned its eponymous gold standard [76,77]. The beneficial role of mouthwashes containing cetylpyridinium chloride in the prevention of tooth decay, has also been demonstrated [78,79]. The antibacterial effect of Listerine has also been shown in both in vitro and in vivo studies [80,81].

Flossing is an alternative to plaque removal, but it is carried out by only 33.42% of subjects and only 10.3% of them do this consistently. The results obtained are in accordance with the data obtained in certain studies conducted and published in the literature [48,50,82,83,84,85]. In general, according to specialized studies, regardless of the adjuvants used, when they are used, an improvement in oral hygiene indices, plaque indices and bleeding indices is obtained [86,87,88,89,90,91]. Use of dental floss was not a daily behavior for most teenagers, as indicated by other specialized studies [92,93]. Students should be trained by dental practitioners regarding the use of dental floss, a fact highlighted in other specialized studies [94,95].

This study has some limitations that need to be considered: year of study or specialization was uneven, the subjects were randomly selected and the bias of any analyzed group was not followed.

## 5. Conclusions

Within the limits of this study, we can draw the following conclusions: The testable hypothesis is true, there are different attitudes regarding oral hygiene depending on the level of education. Students have a deficient level of knowledge regarding the principles and rules of oral hygiene promoted through health education. Most students have a minimum, basic knowledge of the frequency of brushing, the purpose and time when it should be carried out. About half of the subjects do not know the recommended amount of toothpaste to be used during brushing. Only a quarter of the subjects take into account the fluoride content when choosing their toothpaste. About half of the subjects do not know a brushing technique and choose their brush according to other criteria than the dental practitioner instructions. About one-third of students do not use mouthwash or dental floss. Education programs are needed to improve students’ level of knowledge and attitudes toward oral hygiene.

## Figures and Tables

**Figure 1 medicina-58-00406-f001:**
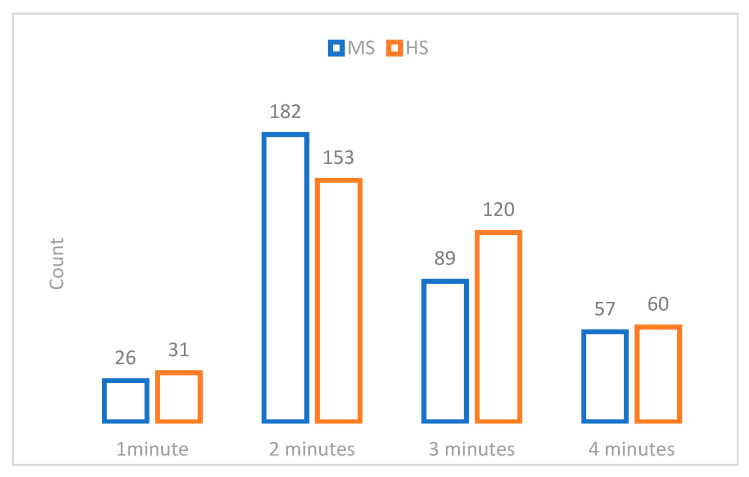
Frequency (count) of time allocated for each dental brushing (count) in function of study level, respectively, MS and HS.

**Table 1 medicina-58-00406-t001:** The questions applied in the questionnaire and the possible answers.

Q 1 = How old are you?
Q 2 = What is your gender? (F = female; M = male).
Q 3 = What is the county where you study? (IS = Iasi; BT = Botosani; SV = Suceava; PH = Prahova; NT = Neamt; B = Bucharest; BC = Bacau).
Q 4 = What is the class level? (MS = middle school/HS = high school).
Q 5 = What class grade are you in? (5-th; 6-th; 7-th; 8-th; 9-th; 10-th; 11-th; 12-th).
Q 6 = Do you know a special dental brushing technique? (Yes/No).
Q 7 = Do you think that dental brushing is done to remove dental bacterial plaque? (Yes/No).
Q 8 = Do you think that dental brushing is done to remove food? (Yes/No).
Q 9 = Do you think that dental brushing is done to have whiter teeth? (Yes/No).
Q 10 = Do you think that dental brushing aims to remineralizers your teeth? (Yes/No).
Q 11 = How long do you think dental brushing should last? 1 = 1 min; 2 = 2 min; 3 = 3 min; 4 = 4 min.
Q 12 = Is the design important when choosing a toothbrush? (Yes/No).
Q 13 = Is the price important when choosing a toothbrush? (Yes/No).
Q 14 = Is the manufacturer company important when buying a toothbrush? (Yes/No).
Q 15 = Is age important when choosing a toothbrush? (Yes/No).
Q 16 = Is it important to follow dental professional recommendations when choosing a toothbrush? (Yes/No).
Q 17 = Is it important to consider advertising when choosing a toothbrush? (Yes/No).
Q 18 = Is it important to take into account the design when choosing toothpaste? (Yes/No).
Q 19 = Is it important to take into account the price when choosing toothpaste? (Yes/No).
Q 20 =Is it important to take the manufacturer company into account when choosing toothpaste? (Yes/No).
Q 21 = Is it important to take into account the fluoride content when choosing toothpaste? (Yes/No).
Q 22 = Is it important to take into account the properties of paste when choosing your toothpaste? (Yes/No).
Q 23 = It is important to take into account advertising when choosing your toothpaste? (Yes/No).
Q 24 = How much toothpaste do you use when you brush your teeth? (1 = the length of the toothbrush; 2 = as much as a pea; 3 = less than a pea).
Q 25 = What is the frequency of brushing? (1 = once a day; 2 = twice a day; 3 = three times a day; 4 = after each meal; 5 = when I feel the need).
Q 26 = When do you brush your teeth? (1 = in the morning; 2 = in the evening; 3 = both morning and evening; 4 = after each meal; 5 = when I feel the need).
Q 27 = Do you use mouthwash? (Yes/No).
Q 28 = When do you use mouthwash? (1 = after each brushing; 2 = when I feel the need; 3 = in the morning; 4 = in the evening; 5 = I don’t use).
Q 29 = Do you floss for dental cleaning? (Yes/No).
Q 30 When do you use dental floss for interdental cleaning? (1 = after each brushing; 2 = when I feel the need; 3 = in the morning; 4 = in the evening; 5 = I do not use).

**Table 2 medicina-58-00406-t002:** Demographic data -distribution of subjects according to class, gender and county.

Count		Study Level		Gender		County							Total
		Ms	Hs	M	F	IS	BT	SV	PH	NT	B	BC	
Class	5th	60	-	21	39	48	0	2	1	0	5	4	60
	6th	116	-	39	77	104	0	7	0	1	3	1	116
	7th	135	-	55	80	38	0	15	2	35	14	31	135
	8th	43	-	20	23	7	0	14	2	16	3	1	43
	9th	-	76	18	58	69	0	3	1	0	3	0	76
	10th	-	86	26	60	59	0	1	19	0	6	1	86
	11th	-	135	38	97	91	2	9	23	1	9	0	135
	12th	-	67	33	34	42	6	5	11	0	3	0	67
Total		354	364	250	468	458	8	56	59	53	46	38	718

**Table 3 medicina-58-00406-t003:** Answers to questions about the criteria used when choosing a toothbrush and toothpaste.

	Questions	Yes	MS	HS	*p*	R
	Criteria for choosing a toothbrush:					
Q12	Is design important?	231	84	147	0 *	−0.178
Q13	Is price important?	140	52	88	0.001 *	0.12
Q14	Is the manufacturer company important?	213	82	131	0 *	−0.14
Q15	Is age important?	247	160	87	0 *	0.224
Q16	Are dental professional recommendations important?	372	202	170	0.003 *	0.104
Q17	Is advertising important?	45	18	27	0.128	−0.048
Criteria for choosing a tube of toothpaste:						
Q18	Is design important?	66	33	33	0.341	0.02
Q19	Is price important?	107	49	58	0.431	−0.029
Q20	Is the manufacturer company important?	220	92	128	0.005 *	−0.1
Q21	Is the fluoride content important?	173	83	90	0.689	−0.015
Q22	Are the properties important?	550	275	275	0.282	0.025
Q23	Is advertising important?	55	20	35	0.031 *	−0.074

* Significance level *p* ≤ 0.05.

**Table 4 medicina-58-00406-t004:** Answers to questions about the attitude towards the amount of toothpaste used, the frequency of toothbrushing use.

	Questions	MS	HS	Total	*p*	R
Q24	How much toothpaste do you use when brushing your teeth?				0.625	−0.035
	1 = the length of the toothbrush	178	195	373		
	2 = as much as a pea	173	167	340		
	3 = less than a pea	3	2	5		
Q25	What is the frequency of brushing?				0.079	0.037
	1 = once a day	55	37	92		
	2 = twice a day	234	248	482		
	3 = three times a day	21	36	57		
	4 = after each meal	25	28	53		
	5 = when I feel the need	19	15	34		
Q26	When do you brush your teeth?				0.099	0.095
	1 = in the morning	36	24	60		
	2 = in the evening	25	15	40		
	3 = both morning and evening	246	262	508		
	4 = after each meal	33	40	73		
	5 = when I feel the need	11	21	32		
	6 = I don’t use	3	2	5		

**Table 5 medicina-58-00406-t005:** Data on the attitude towards the mouthwash and dental floss.

	Questions	MS	HS	Total	*p*	R
Q27	Do you use mouthwash?				0.194	−0.048
	Yes	199	222	421		
	No	155	142	297		
Q28	When do you use mouthwash?				0.062	−0.078
	1 = after each brushing	109	119	228		
	2 = when I feel the need	56	83	139		
	3 = in the morning	20	22	42		
	4 = in the evening	19	12	31		
	5 = I don’t use	150	128	278		
Q29	Do you floss for dental cleaning?				0.051	−0.073
	Yes	106	134	240		
	No	248	230	478		
Q30	When do you use dental floss for interdental cleaning?				0.091	−0.038
	1 = after each brushing	41	33	74		
	2 = when I feel the need	52	80	132		
	3 = in the morning	1	0	1		
	4 = in the evening	19	21	40		
	5 = I do not use	241	230	471		

## Data Availability

The data that support the findings of this study are available on request from the corresponding author.

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
