# Peer review of "Title Cross-Sectional Study to Evaluate Knowledge and Attitudes on Oral Hygiene of Romanian Students"

_medicina, 2022, doi:10.3390/medicina58030406_

Round 1

Reviewer 1 Report

The manuscript has been improved. 

Author Response

REPONSES TO REVIEWER 1

Dear reviewer,

Thank you for your evaluation.

I did the correction in English with and native language speaker.

With thanks and respect,

Iulia Saveanu

Reviewer 2 Report

All my questions were answered and explained clearly. Thanks!

Author Response

REPONSES TO REVIEWER 2

Dear Reviewer,

Thank you for your evaluation.

I completed the introduction. I checked and corrected the material and method section. I did the correction in English with and native language speaker.

With thanks and respect,

Iulia Saveanu

Reviewer 3 Report

Title: Cross-sectional study to evaluate the attitude on oral hygiene

habits of Romanian teenagers

MDPI_medicina_ 1626518- Reviewed 26th february 2022.

REVIEWER:

The authors present a resubmitted manuscript of a cross-sectional study to evaluate the level of knowledge and attitude towards oral hygiene of Romanian students between 10-19 years old, and followed the main ethical issues.

Minor editing changes:

Line 134 – Remove #table 2# from the subchapter description.

Line 249 – Remove #table 3# from the subchapter description.

Line 291 – remove #table 4# from the subchapter description

Line311- Remove #figure 1# from the subchapter description

Line 277- Figure – should be removed

Line 360- Remove #table 5 # from the subchapter description

Author Response

REPONSES TO REVIEWER 3

Dear Reviewer,

Thank you for your evaluation.

I completed the introduction. I checked and corrected the material and method section. I did the correction in English with and native language speaker. I made the requested changes as well.

Line 134 – I remove #table 2# from the subchapter description.

Line 249 – I remove #table 3# from the subchapter description.

Line 291 – I remove #table 4# from the subchapter description

Line311- I remove #figure 1# from the subchapter description

Line 277- Figure – should be removed

Line 360- I remove #table 5 # from the subchapter description

With thanks and respect,

Iulia Saveanu
